# Exosomal microRNAs from Longitudinal Liquid Biopsies for the Prediction of Response to Induction Chemotherapy in High-Risk Neuroblastoma Patients: A Proof of Concept SIOPEN Study ^‖^

**DOI:** 10.3390/cancers11101476

**Published:** 2019-09-30

**Authors:** Martina Morini, Davide Cangelosi, Daniela Segalerba, Danilo Marimpietri, Federica Raggi, Aurora Castellano, Doriana Fruci, Jaime Font de Mora, Adela Cañete, Yania Yáñez, Virginie Viprey, Maria Valeria Corrias, Barbara Carlini, Annalisa Pezzolo, Gudrun Schleiermacher, Katia Mazzocco, Ruth Ladenstein, Angela Rita Sementa, Massimo Conte, Alberto Garaventa, Susan Burchill, Roberto Luksch, Maria Carla Bosco, Alessandra Eva, Luigi Varesio

**Affiliations:** 1Laboratorio di Biologia Molecolare, IRCCS Istituto Giannina Gaslini, 16147 Genova, Italy; davidecangelosi@gaslini.org (D.C.); danielasegalerba@gaslini.org (D.S.); federicaraggi@gaslini.org (F.R.); alessandraeva@gaslini.org (A.E.); luigivaresio@gaslini.org (L.V.); 2U.O.C. Laboratorio di Cellule Staminali Post Natali e Terapie Cellulari, IRCCS Istituto Giannina Gaslini, 16147 Genova, Italy; danilomarimpietri@gaslini.org (D.M.); annalisapezzolo@gaslini.org (A.P.); 3Dipartimento di Oncoematologia, Ospedale Pediatrico Bambino Gesù IRCCS, 00165 Roma, Italy; aurora.castellano@opbg.net (A.C.); doriana.fruci@opbg.net (D.F.); 4Hospital Universitario y Politècnico La Fe, 46026 Valencia, Spain; jaime.fontdemora@gmail.com (J.F.d.M.); canyete_ade@gva.es (A.C.); yanyez_yan@gva.es (Y.Y.); 5Leeds Institute of Medical Research at St James’s University Hospital, Leeds LS9 7TF, UK; V.F.Viprey@leeds.ac.uk (V.V.); s.a.burchill@leeds.ac.uk (S.B.); 6Laboratorio Terapie Sperimentali in Oncologia, IRCCS Istituto Giannina Gaslini, 16147 Genova, Italy; mariavaleriacorrias@gaslini.org; 7Anatomia Patologica, IRCCS Istituto Giannina Gaslini, 16147 Genova, Italy; barbaracarlini@gaslini.org (B.C.); angelaritasementa@gaslini.org (A.R.S.); katiamazzocco@gaslini.org (K.M.); 8Curie Institute, 75248 Paris, France; gudrun.schleiermacher@curie.fr; 9St. Anna Children’s Hospital and Children’s Cancer Research Institute (CCRI), Department of Paediatrics, Medical University, 1090 Vienna, Austria; ruth.ladenstein@ccri.at; 10Oncoematologia Pediatrica, IRCCS Istituto Giannina Gaslini, 16147 Genova, Italy; massimoconte@gaslini.org (M.C.); albertogaraventa@gaslini.org (A.G.); 11Fondazione IRCSS Istituto Nazionale dei Tumori, 20133 Milano, Italy; roberto.luksch@istitutotumori.mi.it

**Keywords:** neuroblastoma, exosomes, chemotherapy response, miRNA, liquid biopsy

## Abstract

Despite intensive treatment, 50% of children with high-risk neuroblastoma (HR-NB) succumb to their disease. Progression through current trials evaluating the efficacy of new treatments for children with HR disease usually depends on an inadequate response to induction chemotherapy, assessed using imaging modalities. In this study, we sought to identify circulating biomarkers that might be detected in a simple blood sample to predict patient response to induction chemotherapy. Since exosomes released by tumor cells can drive tumor growth and chemoresistance, we tested the hypothesis that exosomal microRNA (exo-miRNAs) in blood might predict response to induction chemotherapy. The exo-miRNAs expression profile in plasma samples collected from children treated in HR-NBL-1/SIOPEN before and after induction chemotherapy was compared to identify a three exo-miRs signature that could discriminate between poor and good responders. Exo-miRNAs expression also provided a chemoresistance index predicting the good or poor prognosis of HR-NB patients.

## 1. Introduction

Neuroblastoma (NB) is the most common pediatric extracranial solid tumor, with an incidence of 10 cases per million individuals, and accounting for about 15% of childhood cancer-related deaths. NB arises in the developing sympathetic nervous system and is characterized by high clinical heterogeneity. Based on the International Neuroblastoma Staging System (INSS) patients can be stratified in different risk groups (1, 2, 3, 4, and 4S) ranging from spontaneously regressing to highly malignant and metastatic tumors [1].

Half of all NB patients present a high-risk (HR) disease associated with the presence of aggressive genetic features (*MYCN* amplification, recurrent segmental chromosome aberrations including losses of chromosome 1p, 3p, 4p, 11q and gains of 1q, 2p, 17q) that together with age and stage are currently employed to define the risk level and treatment burden. Despite multi-modality therapy, almost 50% of HR-NB patients suffer from a refractory or relapsed disease [1].

The front-line induction chemotherapy is crucial to reduce tumor burden before surgery and to proceed to consolidation and maintenance treatment. Nowadays, the imaging evaluation is routinely employed to assess patient response to induction chemotherapy. However, biological parameters enabling a dynamic early prediction of inadequate response, which would justify upfront different therapy, are lacking.

The evident need of novel diagnostic molecular tools in oncology has recently led to increasing interest in liquid biopsies as a source of biomarkers, as they provide a minimally invasive method, entailing lower costs and minimizing the invasiveness of tissue sampling [2]. Serial liquid biopsies allow studying the dynamic evolution of the tumor molecular profile, reflecting tumor heterogeneity, evaluating the divergence of different metastatic lesions and the effects caused by the therapeutic stress exerted on tumor cells, and monitoring clonal evolution to better understand the mechanisms of resistance in refractory tumors [2].

Body fluids are a source of exosomes, nanosized extracellular vesicles, ranging from 30 to 120 nm in diameter, that originate from cellular multivesicular bodies (MVBs) as intraluminal vesicles. Exosomes exert important functions in the maintenance of the correct physiological state of the cell, but they have been shown to play a role in pathological conditions, promoting tumor initiation, progression, and metastasis [3].

The exosome content is heterogeneous, comprising proteins, DNA, messenger RNA, microRNAs (miRNAs), other noncoding RNAs, and different lipids that compose the plasma membrane. In particular, miRNAs have been broadly described as biomarkers in different pathologies. Recently, it has been shown that the level of specific miRNAs in serum samples of HR-NB patients was associated with disease burden and treatment response [4].

Because of exosome involvement in determining cancer progression and treatment response, we were interested to investigate exosomal (exo)-miRNAs modulation in liquid biopsies from HR-NB patients after frontline induction chemotherapy to assess the presence of early markers of patient response to treatment. We report here the identification of an exo-miRNAs signature that could discriminate between children with HR-NB who are poor and good responders to induction chemotherapy and predict event-free survival. The exo-miRNAs signature could be further investigated in an early time point to assess whether this approach could be useful to quickly identify patients resistant to the drugs employed in induction chemotherapy and in need of a different drug combination. The present exploratory study demonstrates that analysis of exo-miRNAs modulation is feasible in a large cooperative survey and lays the groundwork for a robust prospective study as part of the HR-NB clinical study.

## 2. Results

### 2.1. Characterization of Exosomes Isolated from Plasma of HR-NB Patients

Exosomes were purified from peripheral blood samples obtained from 52 HR-NB patients whose clinical characteristics are summarized in Table 1. The majority of patients had a stage 4 disease (*n* = 47), 4 subjects had a stage 3 disease, and one patient had a 4S stage disease, according to INSS (International Neuroblastoma Staging System) classification system. MYCN oncogene was amplified in 22 patients, non-amplified in 25 cases, and not evaluable in 5 patients. According to the INRC (International Neuroblastoma Response Criteria) classification system, 6 patients reported a minor response (MR) to induction chemotherapy, 8 patients showed a very good partial response (VGPR), and 33 patients had a partial response (PR). For 5 patients, the data of response to treatment is missing because it was not updated in the SIOPEN database, even if the patients completed the induction chemotherapy. A relapse occurred in 22 patients, while 30 patients did not report any progression or relapse. Finally, 40 patients had a favorable overall outcome while 12 patients deceased.

To investigate the presence of NB-derived exosomes, vesicles collected before and after induction chemotherapy were characterized in terms of size, positivity for the GD2 marker, which is specifically expressed on NB tumor cell surface and barely in normal tissues [1], and miRNAs content. As determined by dynamic light scattering, isolated vesicles had the typical size range of exosomes, comprised between 30 and 120 nm (Figure 1A). Exosomes were further analyzed by flow cytometry in a subset of NB patients (Onset *n* = 9; End of Induction *n* = 9) using a monoclonal antibody (mAb) recognizing the typical exosomal surface marker, tetraspanin CD9 [5], to confirm their purity, and with the NB-specific anti-GD2 mAb. This subset of patients was chosen based on material availability, as this analysis was performed only to determine the presence of tumor-derived exosomes in plasma samples and, thus, the feasibility of the subsequent evaluation studies. The results indicated that about 85–90% of exosomes expressed CD9 both at the onset and at the end of induction chemotherapy, while the mean percentage of GD2^+^ exosomes corresponded to 50% of the whole exosome population in plasma samples at diagnosis and to 27% after the induction treatment (Figure 1B). The difference in the amount of GD2^+^ exosomes observed before and after treatment was statistically significant, as determined by Wilcoxon signed rank test (*p* value < 0.01). The results demonstrate that plasma samples of HR-NB patients contain NB-derived exosomes, which decreased upon induction chemotherapy.

### 2.2. Differential Plasma exo-miRNAs Expression Profile in HR-NB Patients before and after Induction Chemotherapy

To investigate exo-miRNAs modulation upon treatment, exo-miRNAs expression profile was compared in HR-NB patients at diagnosis and after induction chemotherapy. We extracted total RNA from the whole population of isolated exosomes and evaluated the content of miRNAs by capillary electrophoresis to ensure a sufficient retrieval of material for the following analyses. The whole population of circulating exosomes is representative of the tumor status, as cancer cells produce a higher number of exosomes than normal proliferating cells [6]. Tumor-derived exosomes are responsible for reprogramming nonmalignant cells in the tumor microenvironment, which, in turn, could produce exosomes supporting tumor growth and chemoresistance [7]. Thus, the analysis of the whole plasma exosome population may provide a better understanding of tumor response to treatment and of the potential involvement of the tumor microenvironment in chemoresistance or chemosensitivity conditions. The difference between a suitable and an inadequate exo-miRNAs profile is shown in Figure 2A: a significant exo-miRNAs fraction is represented by different peaks in the region between 10 and 50 nucleotides of the small RNA profile (left panel), whereas an inadequate sample is indicated by a flat profile in that region (right panel). About 8% of RNA samples failed the quality control, showing a flat profile, and were discarded from subsequent analyses. Exosomal RNA samples were reverse transcribed, preamplified, and analyzed by human microRNA array cards that allowed to measure the expression of 381 targets for each sample through quantitative real-time PCR (RTqPCR). The raw data used for the analysis are reported in Appendix A. We performed a statistical analysis of the RTqPCR expression data. The absence of a generally recognized approach for exo-miRNAs analysis raised the problem of data processing. RNU6B (U6), a small nucleolar RNA molecule, is currently used as reference gene for miRNAs expression in tumor samples, but it was demonstrated to be unstable in plasma samples [8]. Hence, we set up a new bioinformatics pipeline to be applied in our analysis. The raw data were normalized according to the Global Mean Method [9]. Comparison of exo-miRs expression profiles demonstrated that the number of detected exo-miRNAs was higher in samples at the onset than in those at the end of the treatment (Figure 2B).

We then applied a filter based on the number of undetermined (NA) values. Dealing with such values is challenging when interpreting RTqPCR data because missing data can adversely affect downstream analysis [10]. Imputation methods that substitute NAs with defined values can be included to overcome this when analyzing data. Nevertheless, imputing a high number of NAs could introduce substantial biases, compromising the interpretation of data. To minimize any bias introduced by imputation, only exo-miRNAs detected in at least 80% of samples were included in the analysis. After filtering and imputing the remaining NA values (see Material and Methods), we obtained a total number of 81 exo-miRNAs to be considered for further analysis (Appendix A).

Technical sources of variation, named batch effects, can be introduced in the data when handling and processing samples. Because plasma samples used in our study were collected from different UE centers (see Material and Methods), we wondered whether the different origin of samples could introduce additional technical variations influencing the results. To test this hypothesis, we performed a Principal Variance Component Analysis (PVCA). PVCA pointed out that the contribution of the origin to variability is negligible, accounting for the 3% of the variance, as can be observed in the PVCA bar chart (Appendix A) (see Material and Methods). The Principal Component Analysis (PCA) graph (Appendix A) shows that patients are not differentiated according to their origin; the resulting clusters are overlapping and the origin of samples does not clearly identify different groups of subjects. Usually, a weighted average proportion variance higher than 5% is considered a significant batch effect [11], therefore, we did not apply any algorithm to adjust for the batch effect caused by the origin of samples. Data were then elaborated according to the ΔCT method, calculated as the difference between the ΔCT at the onset and the ΔCT at the end of induction chemotherapy. Differential expression analysis assessed the significance of the modulation of exo-miRNAs expression induced by chemotherapy. Analysis identified 62 exo-miRNAs that were significantly downregulated (fold change, FC > 1.5; *p* value < 0.05) after induction chemotherapy. The list of downregulated exo-miRNAs is reported in Table 2. The analysis showed that induction chemotherapy had a significant inhibitory effect on exo-miRNAs expression, raising the question of the role of exo-miRNAs in patient response to induction chemotherapy. Although the reduced number of detected exo-miRNAs and their downregulation upon treatment could be associated with the observed reduction of GD2^+^ exosomes, the results show that there are tumor-specific exo-miRNA whose modulation is indicative of the response to treatment, as discussed in the next sections.

### 2.3. A Three-miRNAs Signature Differentiates Poor Responders from Good Responders

To determine whether differences in the expression of exo-miRNAs before and after induction chemotherapy could discriminate among patients with different clinical responses, we compared two groups of patients showing clearly different clinical response to the treatment according to the INRC (International Neuroblastoma Response Criteria) classification system [12]. Specifically, we compared patients who showed a poor response, defined as minor response (MR) (n = 6), and patients who showed good response, defined as very good partial response (VGPR) (n = 8). We applied generalized linear model (GLMNET) algorithm [13] to identify the exo-miRs that provided the highest contribution to the separation of MR and VGPR patients. Analysis identified three exo-miRNAs (miR-29c, miR-342-3p and let-7b) that could effectively separate MR from VGPR patients. Our results showed that all the three exo-miRNAs were significantly downregulated after treatment in MR patients, whereas remained almost unchanged in VGPR patients (Figure 3A). In particular, miR-29c showed the highest downregulation in unresponsive patients, with a median FC value of −6.52 compared to VGPR median FC value of −1.57. Let-7b had a median FC value of −3.31 in MR patients and 0.005 in VGPR patients, while miR-342-3p showed a median FC value of −2.83 in MR patients against a median value of −1.19 in VGPR patients. The three-exo-miRNAs signature was validated by RTqPCR. We compared the ΔCT between the two time points (onset and end of chemotherapy) obtained with the Array Card and with RTqPCR to confirm exo-miRNAs differential expression. We found concordance between the two analyses with respect to the three exo-miRNAs differential expression at the end of the induction treatment in the majority of patients (Appendix A).

### 2.4. The Three-miR Signature Depicts Tumor Growth and Chemoresistance

Having established that the exo-miRNAs signature discriminated between MR and VGPR patients, we evaluated whether the three exo-miRNAs could predict pathways involved in the chemotherapeutic response. To this aim, we performed pathway analysis of the exo-miRs signature with miRNet. We found that the three exo-miRNAs are involved in pathways that play a major role in cancer development and chemoresistance. The main enriched pathways known to contribute to tumor progression and survival are reported in Figure 3B. Among them we observed the FoxO signaling pathway, which is involved in both treatment response and resistance acquisition [14], and the PI3K-Akt signaling pathway, which exerts relevant functions in cell proliferation and can be triggered both by neutrophins, growth factors required for the development of sympathetic neurons [15], and by integrins, components of focal adhesion structures mediating the signals between extracellular matrix (ECM) and interacting cells [16]. Interestingly, both the neutrophin and focal adhesion signaling pathways resulted significantly enriched in our analysis (*p* value < 0.05). Enriched pathways included also adherens and gap junctions that, when deregulated, promote metastasis and also the p53 signaling pathway, which is known to play a major role in cell cycle regulation and tumor chemoresistance [17].

### 2.5. Chemoresistance Index Based on exo-miRNAs Modulation after Induction Chemotherapy is Indicative of Event-Free Survival (EFS)

We assessed the efficacy of the three exo-miRNA signature in differentiating PR patients into two clusters that could point out PR subjects closer to MR or VGPR patients. The results of this evaluation are reported in Appendix A, where the PCA plot shows that the three exo-miRNA signature was not able to provide a clear separation of PR patients into two different groups of response to treatment. We also assessed whether the three exo-miRNA signature could predict EFS in the whole cohort of 52 patients. The results of the analysis (Appendix A) showed that each exo-miRNA was not able to significantly predict EFS (Mann–Whitney test > 0.05). Even though the 3 exo-miRNA signature is effective in discriminating between MR and VGPR patients, its efficacy is limited when we include PR patients or we consider EFS. Thus, to better define, at a molecular level, the induction chemotherapy response of PR patients, we established a new approach. We evaluated in the whole cohort of patients (n = 52) all the exo-miRNAs modulated after the induction chemotherapy that have been previously reported to be associated with the response to each chemotherapeutic drug employed in the induction treatment (Table 3) [18,19,20,21,22,23,24,25,26,27,28,29,30,31,32,33,34,35,36,37,38,39,40,41,42,43,44,45,46,47,48,49,50,51,52,53,54,55,56,57,58]. HR-NB patients are treated with a combination of different drugs including cisplatin (CDDP), carboplatin (CBDCA), etoposide (VP-16), doxorubicin (DOXO), vincristine (VCR), and cyclophosphamide (CPM), administered in several cycles (see Material and Methods). Table 4 shows the number of exo-miRNAs known from the literature to be modulated upon treatment and to contribute to chemoresistance to each single drug when overexpressed or downregulated. The literature review included results obtained in different neoplastic diseases, among which NB, and performed mainly on cell line models and only in a few cases on ex vivo human specimens, including blood samples. No involvement of miRNAs in response to CPM has been reported. We calculated a chemoresistance index (CI) for each single patient toward each drug by dividing the number of exo-miRNAs modulated in the patients according to Table 3 by the total number of exo-miRNAs associated with the response to the specific chemotherapeutic drug shown in Table 4 (see Material and Methods).

This targeted approach allowed to estimate the probability of chemoresistance for each patient toward each chemotherapeutic drug used in the induction protocol. The resulting chemoresistance matrix was analyzed by unsupervised k-means clustering to identify homogeneous subgroups of response (Figure 4). We performed the analysis of association between the two clusters identified on the basis of CI values and INSS stage, induction chemotherapy response, MYCN status, or EFS. Only EFS resulted significantly associated with the two clusters (*p* value < 0.05). Cluster 1 showed high CI toward CBDCA, CDDP, and DOXO and was associated with poor responders, while Cluster 2, showing low CI for the same drugs, was associated with good responders. Additionally, we found that Cluster 1 patients were more sensitive to VP-16 and VCR, while cluster 2 patients showed slightly high CI toward these drugs. CBDCA, CDDP, and DOXO seem to have a major impact on the response to treatment, as high CI for these drugs is sufficient to determine worse prognosis, despite the sensitivity/resistance toward VP-16 and VCR. Log-rank test further pointed out the significant difference in survival between the two clusters identified by the CI (*p* value ≤ 0.05), strengthening its predicting value of EFS and treatment response.

## 3. Discussion

HR-NB patients do not respond adequately in around 50% of cases to induction therapy, making the exploration of new and more effective strategies for improving risk stratification at diagnosis crucial for the amelioration of the outcome [2]. In the present study, we investigated the potential use of liquid biopsies to predict HR-NB patient response to induction chemotherapy. We specifically studied exosomes derived from peripheral blood of HR-NB patients, comparing the expression profile of exo-miRNAs at diagnosis and after induction chemotherapy to determine whether exo-miRNAs differential expression can be indicative of tumor response to treatment. We show that a significant percentage of NB-derived exosomes is present in plasma specimens at diagnosis and it is significantly reduced after the chemotherapeutic treatment.

The whole fraction of purified nanovesicles included a significant number of NB-derived exosomes, suggesting that NBs can release exosomes in vivo in agreement with what previously observed in in vitro culture [59] and enabling the analysis of exo-miRNAs derived specifically from the tumor. The reduction of NB-derived exosomes observed after chemotherapy suggests that treatment has an inhibitory effect on the number of exosomes released by the tumor, confirming previous data in ovarian carcinoma [60].

Evaluation of exo-miRNAs expression was challenging because a standardized RTqPCR analysis approach applied to exo-miRNAs is currently lacking. We defined a novel bioinformatics pipeline, named ExoPIPE (see Material and Methods), specifically designed to analyze exo-miRNAs expression. Importantly, we observed that the different origin of samples from several countries had a negligible impact on data variability, preventing the need of removing the batch effects introduced by processing plasma samples.

We identified a signature of three differentially expressed exo-miRNAs (miR-342-3p, -29c, let-7b) that could significantly discriminate between VGPR and MR patients. In particular, MR patients showed a considerable downregulation of miR-342-3p, miR-29c, and let-7b upon treatment, whereas in VGPR subjects their expression remained mainly unchanged. Two patients, classified as VGPR by the INRC system, fell into the group of MR patients according to the 3 exo-miRNAs profile. Interestingly, one of these patients relapsed this year, supporting the efficacy of exo-miRNAs signature in determining patient outcome respect to the INRC system. Two of these miRNAs (miR-29c and let-7b) target MYCN gene [61]. The three exo-miRs have been previously reported to be involved in tumor suppression and chemoresistance. MiR-342-3p was shown to exert tumor-suppressor activity in human cervical cancer by inducing apoptosis and inhibiting cell growth, invasion, and migration [62], and its downregulation caused by glucose starvation and hypoxic conditions has been associated to chemoresistance in triple-negative breast cancer [63]. Hypoxia is a condition of low oxygen tension occurring in solid tumor microenvironment, associated with malignant tumor aggressiveness, and we have previously shown it to be an independent prognostic factor in NB, negatively correlating with tumor outcome [64]. MiR-29c is also a tumor-suppressor miRNA: its expression enhanced the sensitivity of non-small cell lung cancer cells to cisplatin by targeting the PI3K/Akt pathway [65]. Let-7b belongs to the let-7 family, a group of tumor suppressor miRNAs, implicated in numerous cancers, including NB. Let-7 miRNAs directly bind MYCN, inhibiting its expression [66]. LIN28B protein was shown to be an independent risk factor determining a poor outcome in NB, through negative regulation of let-7 miRNA, that leads to a high MYCN protein expression and, therefore, to an aggressive tumor phenotype [67]. Moreover, let-7b downregulation contributes to cisplatin resistance in glioblastoma cell lines promoting tumor cell growth through the upregulation of cyclin D1 [68]. Downregulation of miR-342-3p, let-7b, and miR-29c could thus represent an important mechanism mediating the poor response of HR-NB patients to induction chemotherapy. This represents the first evidence that the analysis of exo-miRs expression profile could be indicative of patient response to induction chemotherapy, according to the clinical definition of treatment outcome. We also tested the hypothesis that the three exo-miRNAs signature discriminating MR and VGPR patients could identify subgroups of patient response in the subset of partial responders (PR), but we were not able to obtain a clear separation. We also observed that the three exo-miRNA signature did not predict EFS in the whole cohort of patients.

Therefore, to perform an analysis considering PR patients and to better define a good or poor chemotherapeutic response, we set up a new approach able to predict treatment outcome at a molecular level. We considered all the exo-miRNAs that were modulated after chemotherapy in the cohort of 52 HR-NB patients for which up or downregulation was shown in the literature to be associated with resistance toward the drugs employed in the induction protocol. Then, we measured exo-miRNAs expression levels in response to induction chemotherapy in each patient of our cohort. We calculated the chemoresistance index (CI) on the basis of the number of exo-miRNAs, related to a specific drug, that were modulated according to a chemoresistant behavior (see Table 3). A smoothing method was applied to avoid potential biases caused by the low number of miRNAs associated with specific drug resistance. In this way, we were able to determine the CI for each single NB patient toward each single drug, limiting the effects of potential artifacts. We used this information to calculate the CI for each single NB patient toward each single drug. The *CI* was able to effectively stratify HR-NB patients into two groups that were significantly associated with a poor and a good response on the bases of EFS, calculated with a 3-year follow up. Poor responders showed high *CI* for CBDCA, CDDP, and DOXO. High resistance toward these drugs appeared to be sufficient to determine a poor treatment outcome and unfavorable prognosis, as the sensitivity to VP-16 and VCR was not able to overcome CBDCA, CDDP, and DOXO resistance, while good responders showed exactly the opposite CI. Survival analysis confirmed the results. Our cohort included patients with at most 3 years follow-up, and the association analysis would certainly be more robust if we could have had available patients with at least 5 years follow-up. The present exploratory study, however, indicates that the CI calculated on the basis of exo-miRNA modulation could define patient response to specific drugs at a molecular level. The novelty of the CI makes it an important tool in personalized medicine but, at the same time, should be carefully validated in order to define a reliable parameter for tumor chemoresistance evaluation. The CI would improve and complete the information obtained with the three exo-miRNA signature, which can be applied only to a subset of patients. If our results are confirmed, a panel of exo-miRNAs could be customized as an array to assess the response to chemotherapy for each patient. In particular, the exo-miRNA evaluation performed in the middle phase of induction chemotherapy would allow to adjust the treatment according to the specific patient sensitivity/resistance profile. Therefore, both the three exo-miRNA signature and CI could be useful tools for the clinician in the decision-making process to define which drug combination apply for every single patient.

Since the present study suffers from an important bias due to the limited cohort of patients analyzed, our proposal is to validate the prognostic value of exo-miRNAs here identified in a prospective study on HR-NB patients who will be enrolled in the next HR-NB study protocol.

## 4. Materials and Methods

### 4.1. Study Population and Blood Sample Collection

Patients affected by HR-NB enrolled in the HR-NBL Study of SIOP-Europe (ClinicalTrials.gov
*identifier: NCT01704716*) were considered for this study. Patients were recruited at enrollment into the HR-NBL SIOPEN study starting in 2016 without being preselected for the analysis. In this protocol, patients were randomized in the induction phase to receive one of two regimens, Rapid-COJEC (80 days treatment with two courses of carboplatin, etoposide, vincristine; four courses of cisplatin, vincristine; two courses of etoposide, and cyclophosphamide, 1 every 10 days) or modified-N7 regimen (105 days treatment with combinations of vincristine, doxorubicin, cyclophosphamide, cisplatin, and etoposide in 5 courses, 1 every 3 weeks). The procedures were carried out according to the proper guidelines and in adherence with the ethical principles of the Declaration of Helsinki. Written informed consent was provided from the parents or legal guardian of patients enrolled in this study. Blood samples from 52 patients were collected in ACD or EDTA tubes at two time points: before and at the end of the induction chemotherapy, for a total number of 104 samples analyzed. Most blood samples were provided by the Italian AIEOP (Associazione Italiana Emato-Oncologia Pediatrica) centers. Further, frozen plasma samples were from La Fe Hospital (Valencia, Spain), the Leeds Institute of Medical Research (Leeds, UK), and the Curie Institute (Paris, France). Blood was centrifuged at 1200 *g* for 10 min at room temperature (RT) to collect plasma. Plasma was stored at −80 °C or used immediately for exosomes isolation. Blood processing was performed within 24 hours from sampling. The study cohort included mainly patients with a stage 4 disease, with different clinical and biological features, as reported in Table 1.

### 4.2. Exosome Isolation and miRNAs Purification

Exosomes isolation and RNA extraction were performed with the exoRNeasy Serum/Plasma Midi kit (Qiagen Italia, Milan, Italy). Briefly, 500 μL of plasma was centrifuged at 16,000 × *g* and 4 °C to eliminate cellular debris. Supernatants were then mixed with one volume of XBP binding buffer, loaded onto exoEasy spin columns, and centrifuged at 500 × *g* 1 minute at RT. XWP washing buffer was then added (3.5 mL) to the filter column and spun at 5000 × *g* for 5 min at RT. The spin column was transferred to a new collection tube and centrifuged at 5000 × *g* for 5 min at RT after the addition of 700 μL of QIAzol to the membrane to collect the lysate. RNA extraction was then performed according to the manufacturer’s instructions. Quality control and miRNAs evaluation were carried out on the Agilent 2100 Bioanalyzer, using the small RNA assay (Agilent Technologies Spa, Milan, Italy).

### 4.3. Flow Cytometric Analysis

To collect intact exosomes, the exoRNeasy Serum/Plasma Midi kit protocol was used: in the last step of the process, QIAzol was substituted with 200 μL of Buffer XE, a reagent enabling elution and collection of exosomes. Exosome size was evaluated using the zetasizer nano ZS90 particle sizer (Malvern Instruments, Worchestershire, UK). Retrieved exosomes were then analyzed for the presence of CD9 and GD2 markers by flow cytometry after vesicles adsorption onto latex beads, as previously described [69]. Briefly, collected exosomes were incubated with 2 μL of 4 μm diameter aldehyde/sulfate latex beads (Invitrogen, Life Technologies Italia, Monza, Italy) for 2 hours at RT and then incubated for 30 min at RT in PBS supplemented with 2% FBS. Exosomes-coated beads were incubated for 30 min at 4 °C with primary mouse anti-human PE-conjugated monoclonal antibodies (mAbs) to CD9 or GD2. An isotype-matched PE-conjugated primary mAb was used as negative control. All mAbs were used in accordance with the manufacturer instructions. Samples were analyzed by Gallios flow cytometer and Kaluza software (Beckman Coulter, Milano, Italy).

### 4.4. RTqPCR Analysis

Exo-miRNAs were analyzed by the TaqMan Array Card Technology. Briefly, 50 ng of RNA was reverse transcribed with the TaqMan® microRNA Reverse Transcription Kit, using the Megaplex^TM^ RT primers Human Pool A (Thermo Fisher Scientific, Monza, MB, Italy). Pre-amplification of cDNA was performed with TaqMan® PreAmp Master Mix and Megaplex^TM^ Pre-Amp primers Human Pool A. The pre-amplification product was diluted according to the manufacturer’s instructions and used to perform microRNA profiling on the ViiA^TM^ 7 Real-Time PCR System. Briefly, 9 μL of the diluted pre-amplified product was mixed with 450 μL TaqMan® Universal Master Mix II, No UNG (Thermo Fisher Scientific, Monza, MB, Italy), and 441 μL of nuclease-free water. 100 μL of the PCR reaction mix was dispensed into each well of the TaqMan® Array human microRNA A card (Thermo Fisher Scientific, Monza, MB, Italy), enabling the quantification of 381 human miRNAs. The validation was performed with individual qPCR assays based on specific TaqMan miRNAs Assays (Thermo Fisher Scientific, Monza, MB, Italy). Samples were run in triplicate on MicroAmp Fast Optical 96-well reaction plate (Thermo Fisher Scientific, Monza, MB, Italy).

### 4.5. Bioinformatic Procedures and Statistical Analysis

We designed a new bioinformatic pipeline implemented in R, ExoPIPE, for the analysis of exo-miRNAs expression measured by quantitative RTqPCR technology. The pipeline uses functions implemented in the published R packages and R modules implemented in our laboratory. ExoPIPE is composed by data preprocessing, CT filtering, data normalization, data imputation, and feature selection. Any CT value higher than 32, lower than 14, or undergoing an experiment failure was categorized as unreliable, as recommended by the manufacturer guidelines (ViiA 7 Software, Thermo Fisher Scientific). Raw data were normalized by global mean method (GM) [9]. The global mean value of each sample, calculated by considering only the exo-miRNAs expressed with a threshold cycle (CT) lower than 32, was subtracted from the raw CT of each miRNAs (ΔCT). The Global mean normalization method was effective in reducing technical variability, as supported by the cumulative distribution analysis. The coefficient of variation (CV), a parameter to evaluate the removal of technical variability [9], was calculated for both raw and normalized data: lower CV is indicative of a better removal of experimentally induced noise. Indeed, normalized data showed a significantly lower CV compared to raw data (Kolmogorov-Smirnov *p* value < 0.001) (Appendix A). The boxplot in Appendix A further shows that normalization was able to remove the differences due to the experimental noise, thus rendering the median values more similar to each other and data more comparable. NA values were imputed with the lowest expression value of the respective miRNA minus one log2 unit. This results in one CT unit value higher than the highest CT recorded for that miRNA in all samples [10]. The raw data used for the analysis are reported in the Appendix A.

Feature selection to identify relevant exo-miRNAs was performed with GLMNET [13] setting up the clinical response as the response variable, the exo-miRNAs as explanatory variables, elastic net mixing parameter at 0.7 and the minimum mean square error as gamma value. Gamma value was assessed using the leave-one-out cross-validation technique. We assessed the presence of any batch effects using principal variance component analysis (PVCA) [11]. We used the country of origin of the samples as batch variable and the age of patient, the INSS stage, and the MYCN status of the tumor as biological variables. PVCA provided a weighted average proportion variance (WAPV) for each variable and combination of variables. A WAPV of the batch variable lower than 5% was considered a negligible batch effect [11].

Among the exo-miRNAs modulated after the induction chemotherapy, we identified a “chemoresistance list” including miRNAs known from the literature to be up or downregulated in chemoresistance condition for each chemotherapeutic drug used in the NB induction therapy. By looking at our data, we analyzed the modulation after the induction chemotherapy of the exo-miRNAs in the “chemoresistance list” in each patient. The −ΔΔCT values were used to define the up or downregulation upon treatment. We counted the miRNAs coordinately up or downregulated according to the “chemoresistance list” for each patient and each single drug. This number and the total number of miR in the “chemoresistance list” have been used to estimate the probability of chemoresistance of a patient to a chemotherapeutic drug using the Laplace smoothing method. Heat map visualization and unsupervised k-means clustering analysis were carried out using Morpheus: versatile matrix visualization and analysis software (Morpheus, Broad Institute, Cambridge Massachusetts, USA). Z test for proportions was used to estimate whether two patient groups differ significantly on one characteristic. Three-year event-free survival (EFS) curves were compared with the log-rank test using GraphPad Prism version 6.0 for MAC, GraphPad Software, San Diego California USA. *p* values lower than 0.05 were considered significant.

### 4.6. Study Approval

Patients affected by HR-NB enrolled in the HR-NBL Study of SIOP-Europe (ClinicalTrials.gov
*identifier: NCT01704716*) were considered for this study. The study was approved in 2016 by Comitato Etico Regionale (Verbale n. 11/2016). The procedures were carried out according to the proper guidelines and in adherence with the ethical principles of the Declaration of Helsinki. Written informed consent was provided from the parents or legal guardian of patients enrolled in this study.

## 5. Conclusions

Our results demonstrate that the blood of HR-NB patients contains tumor-specific exosomes and that exo-miRNAs can be efficiently isolated from plasma specimens. The possibility of defining patient response on the bases of molecular parameters meets the concept of precision medicine, which enables to distinguish patients with similar clinical presentations but different cellular and molecular responses. In this case, response to induction chemotherapy, currently defined by imaging and clinical parameters, could potentially be determined by blood exo-miRNAs expression. To our knowledge, this is the first study that investigates exosomes in HR-NB plasma samples and identifies exo-miRNAs as indicators of induction chemotherapy response. Our results pave the way for exo-miRNAs application in liquid biopsies as circulating biomarkers of chemotherapeutic response and for the development of NB-targeted treatment.

## Figures and Tables

**Figure 1 cancers-11-01476-f001:**
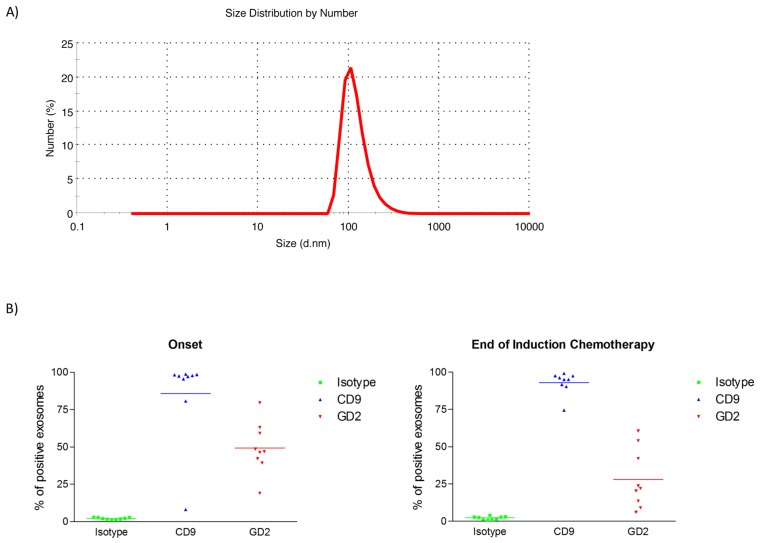
Characterization of exosomes isolated from plasma of high-risk neuroblastoma (HR-NB) patients and of small RNA content. (**A**) Dynamic light scattering analysis shows the size distribution of isolated microvesicles from a representative NB patient. The diameter (nanometers) is reported on X-axis, while the number of microvesicles is on the Y-axis. (**B**) Flow cytometry results show the percentage of CD9^+^ vesicles and GD2^+^ exosomes in a subset of patients before (n = 9) and after (n = 9) induction chemotherapy. The difference in GD2^+^ exosomes observed after treatment is significant (Wilcoxon signed rank test, *p* value < 0.01).

**Figure 2 cancers-11-01476-f002:**
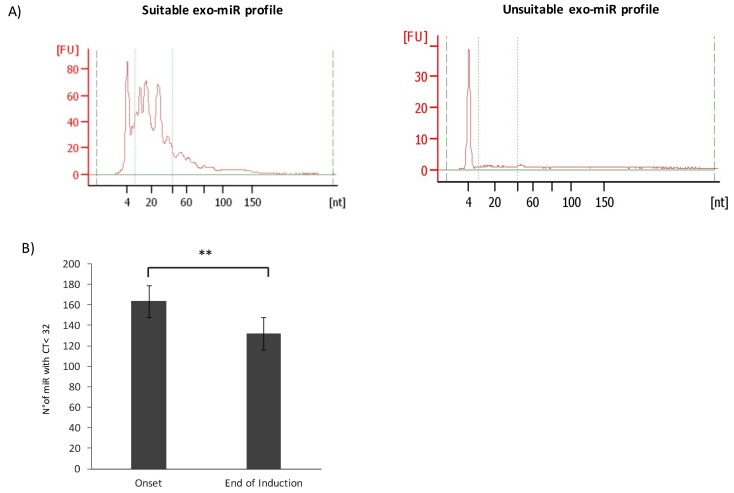
Exosomal (exo)-miRNAs expression in plasma samples. (**A**) Results of capillary electrophoresis, performed with the small RNA assay (Agilent 2100 Bioanalyzer), show the adequate exo-miRNAs profile of a representative NB patient and an unsuitable exo-miRNAs profile. The miRNAs region goes from 10 to 50 nucleotides included between the dotted lines. The Y-axis represents the fluorescence units [FU] and the X-axis reports the length of RNA molecules in nucleotides [nt]. **(B**) The bar chart shows the average number of detected miRNAs (CT value < 32) in plasma exosomes at the onset and at the end of treatment. Error bars represent SD, ** *p* < 0.001.

**Figure 3 cancers-11-01476-f003:**
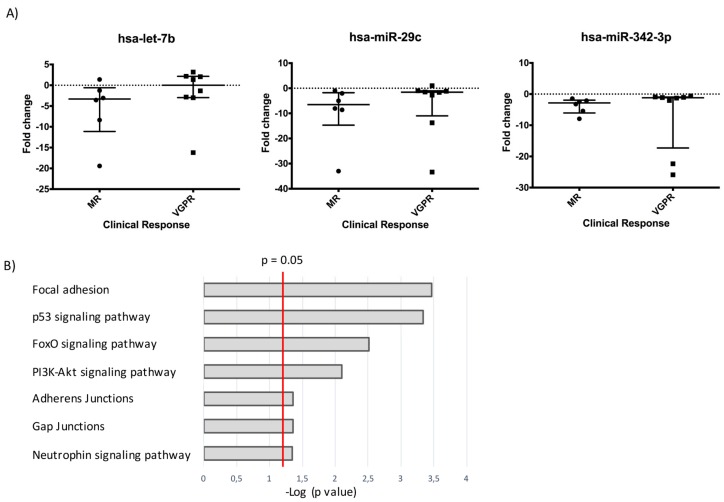
A three exo-miRNAs signature differentiates MR and VGPR patients. (**A**) Dot plot graphs show the FC values after induction chemotherapy of each exo-miRNAs for each MR (minor responders) and VGPR (very good partial responders) patient. The median and the interquantile range values are reported. In VGPR patients let-7b, miR-342-3p, and miR-29c are not modulated, while in MR patients they are downregulated. (**B**) Result of the pathway analysis performed with miRNet showing the enrichment of several pathways related to cancer development and chemoresistance. The bar chart reports the most significantly enriched pathways, listed by decreasing *p* values, that are relevant for tumor progression and survival.

**Figure 4 cancers-11-01476-f004:**
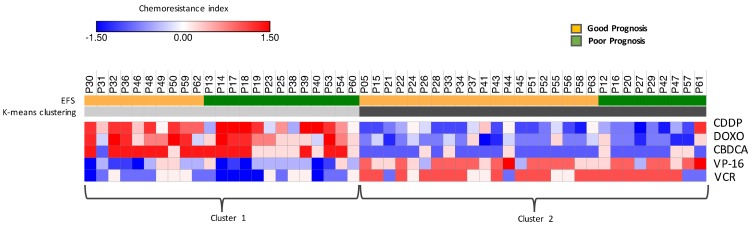
Chemoresistance Index differentiating good and poor responders. K-means clustering built on the CI. The algorithm identifies two groups of patients: cluster 1 includes subjects showing high CI toward CBDCA, CDDP, and DOXO and low CI for VP-16 and VCR; cluster 2 contains patients showing opposite features, with low CI toward CBDCA, CDDP, and DOXO and slightly high CI to VP-16 and VCR. Event-free survival (EFS) defined the response to induction chemotherapy: patients that reported a relapse, a progression, or died are addressed as poor responders. The association between the identified clusters and EFS is significant (*p* value < 0.03), showing that the CI is indicative of a poor response.

**Table 1 cancers-11-01476-t001:** Clinical information of the patient cohort.

Clinical Features	Classification	Patients (*n* = 52)
INSS stage	4	47 (91%)
	3	4 (7%)
	4S	1 (2%)
MYCN status	AMPL	22 (42%)
	NO AMPL	25 (48%)
	NA	5 (10%)
Induction Response	MR	6 (12%)
	PR	33 (63%)
	VGPR	8 (15%)
	NA	5 (10%)
Relapse	YES	22 (43%)
	NO	30 (57%)
Overall Survival	ALIVE	40 (77%)
	DECEASED	12 (23%)

The table reports the main clinical parameters of the patients enrolled in the study: stage of disease according to INSS (International Neuroblastoma Staging System), MYCN oncogene status, clinical response to induction chemotherapy according to INRC classification system, relapse, and overall survival. AMPL = amplified; NO AMPL = not amplified; NA = not available; MR = minor response; PR = partial response; VGPR= very good partial response.

**Table 2 cancers-11-01476-t002:** Differentially expressed exo-miRNAs at the end of induction chemotherapy.

miRNA ID ^a^	Mean ∆CT End	Mean ∆CT Onset	Mean RQ End	Mean RQ Onset	Fold Change ^b^	Adjusted *p* Value ^c^
hsa-miR-376a	3.28	1.28	0.10	0.41	−2.0	5.30E−08
hsa-miR-150	−6.04	−7.60	65.87	194.19	−1.6	9.87E−08
hsa-miR-376c	2.69	0.70	0.16	0.62	−2.0	1.48E−07
hsa-miR-132	1.33	−0.55	0.40	1.47	−1.9	5.45E−07
hsa-miR-130a	1.36	−0.49	0.39	1.40	−1.8	1.21E−05
hsa-miR-539	2.61	0.85	0.16	0.55	−1.8	3.22E−05
hsa-miR-342-3p	−2.59	−3.70	6.03	12.96	−1.1	4.81E−05
hsa-miR-192	1.40	0.06	0.38	0.96	−1.3	5.95E−05
hsa-miR-340	3.02	1.78	0.12	0.29	−1.2	5.95E−05
hsa-miR-320	−4.01	−4.97	16.10	31.28	−1.0	7.29E−05
hsa-let-7g	0.24	−1.06	0.85	2.08	−1.3	9.93E−05
hsa-miR-590-5p	0.96	−0.30	0.51	1.23	−1.3	9.93E−05
hsa-miR-152	2.44	1.22	0.18	0.43	−1.2	2.27E−04
hsa-miR-146b	−3.45	−4.37	10.91	20.74	−0.9	2.27E−04
hsa-miR-20a	−3.86	−5.32	14.56	39.97	−1.5	3.64E−04
hsa-miR-25	−0.31	−1.69	1.24	3.23	−1.4	3.64E−04
hsa-miR-26b	0.19	−1.15	0.88	2.22	−1.3	3.64E−04
hsa-miR-324-3p	1.73	0.62	0.30	0.65	−1.1	4.52E−04
hsa-miR-199a-3p	−0.01	−1.41	1.01	2.65	−1.4	4.95E−04
hsa-miR-106b	−0.59	−1.75	1.50	3.36	−1.2	5.83E−04
hsa-miR-19b	−4.65	−6.15	25.11	71.00	−1.5	6.23E−04
hsa-miR-374	−1.50	−2.61	2.82	6.11	−1.1	6.54E−04
hsa-miR-29c	2.42	1.03	0.19	0.49	−1.4	9.50E−04
hsa-miR-20b	−1.84	−2.92	3.57	7.55	−1.1	1.03E−03
hsa-miR-26a	−1.83	−3.01	3.57	8.07	−1.2	1.09E−03
hsa-miR-140-3p	3.24	2.38	0.11	0.19	−0.9	1.65E−03
hsa-miR-18a	1.47	0.31	0.36	0.80	−1.2	1.65E−03
hsa-miR-185	1.08	−0.04	0.47	1.03	−1.1	1.65E−03
hsa-miR-195	−0.71	−1.81	1.63	3.50	−1.1	1.69E−03
hsa-miR-142-3p	−2.80	−4.16	6.95	17.86	−1.4	2.02E−03
hsa-miR-21	−1.44	−2.66	2.72	6.31	−1.2	2.30E−03
hsa-miR-652	1.07	−0.15	0.47	1.11	−1.2	2.41E−03
hsa-miR-19a	−0.08	−1.40	1.06	2.65	−1.3	2.43E−03
hsa-let-7e	−2.27	−3.31	4.81	9.94	−1.0	2.55E−03
hsa-miR-106a	−5.59	−6.46	48.08	88.20	−0.9	2.55E−03
hsa-miR-660	1.40	0.29	0.38	0.82	−1.1	2.99E−03
hsa-miR-598	1.94	0.64	0.26	0.64	−1.3	3.56E−03
hsa-let-7d	0.11	−0.81	0.93	1.75	−0.9	3.73E−03
hsa-miR-24	−4.99	−5.75	31.85	53.81	−0.8	5.38E−03
hsa-miR-17	−5.42	−6.41	42.81	85.03	−1.0	5.47E−03
hsa-miR-483-5p	−0.30	−1.11	1.23	2.16	−0.8	5.67E−03
hsa-miR-30b	−2.76	−3.76	6.78	13.54	−1.0	5.99E−03
hsa-miR-301	1.70	0.70	0.31	0.62	−1.0	6.28E−03
hsa-miR-186	−2.53	−3.38	5.79	10.40	−0.8	8.42E−03
hsa-miR-331	−1.40	−2.13	2.63	4.38	−0.7	8.98E−03
hsa-miR-27a	0.15	−0.82	0.90	1.76	−1.0	9.19E−03
hsa-miR-15b	−0.63	−1.60	1.55	3.04	−1.0	9.41E−03
hsa-miR-28	2.12	1.28	0.23	0.41	−0.8	1.14E−02
hsa-miR-345	0.70	0.05	0.61	0.96	−0.6	1.28E−02
hsa-miR-30c	−3.10	−4.03	8.59	16.32	−0.9	1.29E−02
hsa-miR-103	0.95	0.13	0.52	0.91	−0.8	1.30E−02
hsa-miR-193b	1.91	1.21	0.27	0.43	−0.7	1.34E−02
hsa-miR-29a	−0.84	−1.54	1.78	2.91	−0.7	1.64E−02
hsa-miR-99b	2.30	1.69	0.20	0.31	−0.6	1.64E−02
hsa-miR-328	0.69	−0.16	0.62	1.12	−0.9	1.80E−02
hsa-miR-744	0.24	−0.53	0.85	1.45	−0.8	2.78E−02
hsa-miR-425-5p	0.38	−0.23	0.77	1.18	−0.6	3.23E−02
hsa-miR-92a	−2.30	−3.03	4.94	8.15	−0.7	3.23E−02
hsa-miR-16	−5.99	−6.86	63.78	116.19	−0.9	3.25E−02
hsa-miR-142-5p	2.39	1.66	0.19	0.32	−0.7	3.38E−02
hsa-miR-221	0.16	−0.73	0.90	1.66	−0.9	4.11E−02
hsa-miR-146a	−5.33	−5.98	40.31	62.91	−0.6	4.30E−02

^a^ List of differentially expressed exo-miRNAs; ^b^ Fold change defined as –∆∆CT value. Fold change lower than 0 indicates a downregulation. Fold change higher than 0.58 or lower than −0.58 was considered significant; ^c^ T test *p* value adjusted for multiple hypothesis testing using Benjamini-Hockberg. *p* value lower than 0.05 is considered significant.

**Table 3 cancers-11-01476-t003:** Exo-miR associated to the response to chemotherapeutic drugs.

miRNA	Cisplatin	Etoposide	Doxorubicin	Vincristine	Carboplatin	Ref.
hsa-miR-150	↑					[18]
hsa-miR-342-3p	↑					[19]
hsa-miR-320	↑					[20]
hsa-let-7g	↓					[21]
hsa-miR-590-5p	↑					[22]
hsa-miR-25	↑					[23]
hsa-miR-20a	↑		↑			[24,25]
hsa-miR-146b	↑		↓	↓		[26]
hsa-miR-106b			↑			[27]
hsa-miR-199a-3p			↓			[28]
hsa-miR-195	↓	↓				[29]
hsa-miR-26a			↓			[30]
hsa-miR-483-5p	↑					[31]
hsa-let-7d	↓		↓			[32]
hsa-miR-106a	↑		↑			[33,34]
hsa-let-7e			↑			[35]
hsa-miR-21	↑	↑	↑			[36,37]
hsa-miR-17	↑					[38]
hsa-miR-15b	↑			↓		[39,40]
hsa-miR-24	↑	↓	↑			[41,42]
hsa-miR-30c	↑		↓			[43,44]
hsa-miR-345	↓					[45]
hsa-miR-29a	↓	↓				[46,47]
hsa-miR-16		↓				[40]
hsa-miR-223			↑		↑	[48,49]
hsa-miR-222	↑		↑		↑	[49,50]
hsa-miR-146a	↓	↓				[51,52]
hsa-let-7b	↓		↓			[53]
hsa-miR-125a-5p	↑					[54]
hsa-miR-155	↑		↑			[55]
hsa-miR-486	↑					[56]
hsa-miR-126				↓		[57]
hsa-miR-191	↑		↑			[58]

The table shows all the exo-miRNAs modulated in the HR-NB cohort analyzed in response to induction therapy that are reported in the literature to contribute to the response to each single drug. The up- or downregulation of each exo-miR promoting chemoresistance is indicated. References are reported in the last column.

**Table 4 cancers-11-01476-t004:** Number of exo-miRNAs associated with the response to chemotherapeutic drugs.

Drug	Chemoresistance	
	N° Upregulated exo-miRNAs	N° Downregulated exo-miRNAs
Cisplatin	19	7
Etoposide	1	5
Doxorubicin	10	6
Vincristine	0	3
Carboplatin	2	0
Cyclophosphamide	0	0

The table shows the number of exo-miRNAs reported in the literature to contribute to the response to each indicated drug. This table has been used as a reference to calculate the chemoresistance index.

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
