# Peer review of "Exosomal microRNAs from Longitudinal Liquid Biopsies for the Prediction of Response to Induction Chemotherapy in High-Risk Neuroblastoma Patients: A Proof of Concept SIOPEN Study"

_cancers, 2019, doi:10.3390/cancers11101476_

Round 1

Reviewer 1 Report

The authors present data about 52 patients with HR NB, about half of whom relapsed, and ¼ of whom died of disease. They investigated changes in NB-derived exosomal miRNAs in response to induction therapy and differences in miRNA profiles between patients who died of disease vs survived. Overall, this is an important study that utilizes a blood-based biomarker to monitor disease, though it’s limited by small sample size. The authors are essentially reporting on a feasibility study that lays the groundwork for a more robust study as part of the current HR NB clinical study. The decrease in exosome numbers is not particularly compelling (Fig 1) and the miRNA expression decrease at end induction is not particularly compelling (Fig 2). The final comparison was 78 exo-miRNAs in a small number of patients (based on quality of data obtained) and the 2-6 fold difference in the 3 miRNAs have an uncertain significance level (not noted in Fig 3).

The review of miRNAs relevant in NB is a nice addition to the paper.

Major critique:

Very limited ability to draw conclusions with exploratory analyses that are may not be replicated. Therefore, the title is overly generous in terms of the findings/conclusions from the paper and it should be modified. This was an exploratory pilot-like study with insufficient sample size to draw a robust conclusion as stated in the title, yet it is still an important proof-of-concept study.

Minor critique:

More has been published about the role of let-7 miRNAs in NB and should therefore be considered to further enhance the depth of discussion.

Reviewer 2 Report

Exosomal-microRNAs from longitudinal liquid biopses predict response to induction-chemotherapy in high-risk neuroblastoma patients: a SIOPEN study, by Morini, Cangelosi et al., presents a very interesting approach that could potentially allow early discrimination of high-risk neuroblastoma patients based on their positive or negative response to chemotherapy, using a fast and lowly invasive method consisting on isolating and analysing exosomal miRNA from blood samples.

The idea is extremely appealing, since high-risk neuroblastoma patients constitute a group of patients with high mortality, in need of the development of new treatment regimes to improve survival and reduce side effects.

I find the study very interesting and well presented, but I suggest that a few points require clarification and/or changes. I will enumerate those points below, following the order in which they appear in the manuscript:

The criteria used to define treatment response as MR, PR, VGPR are not defined till section 2.3, line 189. However, it would be clarifying for the reader to mention this criteria (INRC classification system) already in the legend of Table 1.

In section 2.1, Figure 1, it can be seen that not in all patients the levels of GD2+ exosomes are reduced to the same extent after treatment. Were the patients used for the exosome characterization by flow cytometry good or bad responders? Was there a correlation between the levels of GD2+ exosomes and the response to treatment?

There is a small typo in the legend fo Figure 1A: "Dynamiclight".

In section 2.2, the authors should explain the reason why the exo-miRs expression profile was analysed using the whole population of exosomes, and not only the GD2+ exosomes. Is it due to technical limitations to specifically isolate the tumor-derived exosomes? The authors should additionally explain any possible impact of this point on their results and conclusions. The authors should also discuss the possibility that the reduced number of exo-miRs (Figure 2B), as well as the reduced Fold Change for all analysed miRs (Table 2) after treatment could be due to the reduced number of GD2+ exosomes. Are the data shown in Figure 2B and Table 2 somehow normalised to the number of GD2+ exosomes?

Table 2 does not show the expression change data for hsa-let-7b, which is later on used, together to other two miRs, to define the response miR signature in section 2.3.

Table 2 shows a reduction of the expression of all the analysed miRs after treatment. However, this data is presented as the average of all the patients analysed. Later on, in section 2.3 and Figure 3, the authors argue that the levels of the three miRs selected for the miR signature are not changed in the good responders. Thus, it seems obvious that using the average fold change for all patients in Table 2 is masking relevant information. I suggest to include standard deviation or, even better, include a supplemental table/figure with the information organised patient by patient or at least grouped by good/bad responders.

The terminology for good and poor response should be unified, or an equivalence clearly stated somewhere, between text and figures. For example Table 1 and Figure 3 use MR (minor response), PR (partial response) and VGPR (very good partial response), while in the text in lines 193, 300-302, and others, it is used "poor" and "good responders"..

Has any statistical analysis been performed to check the significance in Figure 3A? At a glance it does not look like the differences between MR and VGPR are very big. The median values are very close, especially in the case of hsa-miR-342-3p, and the dispersion of the data could have a negative impact on a statistical test.

What is exactly the message that the authors want to deliver by using the exo-miR signature (Section 2.4) in relation with the chemo-resistance index (Secton 2.5)? The authors should integrate both ideas in the discussion. Both things have the objective of discriminating good and poor responders. What is the meaning of using both? How do they complement each other? What is the authors' suggestion about the potential use of both approaches in the clinic?

The authors should clarify the criteria used to create the chemoresistance index, and critically discuss its pertinence. As it is now explained in the manuscript, it seems not a very robust choice. The changes on the expression of specific miRs after treatment with each of the chemotherapeutic agents in other cancer cell types (and not even in exosomes) might not necessarily reflect what happens in neuroblastoma patients. Additionally, the simultaneous use of all these drugs during induction therapy for neuroblastoma most likely have an impact on the final outcome of miRs expression, and it might not reflect the effect of individual drugs. As for the miRs considered in Table 4, do they include only miRs evaluated in the present study, or others? The results of applying this chemoresistance index point out two clusters, in which cluster 1 is associated to poor responders, and cluster 2 to good responders. However, 43,5% (10 out of 23) of the patients included in cluster 1 showed good prognosis. That means that almost half of the patients predicted as poor responders by the chemoresistance index were actually good responders. Having all this into account, the authors should consider to redefine the chemoresitance index, or at least very critically discuss its weaknesses and outline and alternative.

Furthermore, the patients used for this study were subjected to two different chemotherapeutic regimes. The authors should address if these different regimes have an impact on their observations. 

It would be interesting to correlate the good and poor responders, according the the miRs profile, to the clinical parameters shown in Table 2.

Reviewer 3 Report

The investigators present an interesting and important analysis of circulating peripheral blood exosomes in subjects with high-risk neuroblastoma and the exosome content of microRNA (miRNA) at diagnosis and after induction of chemotherapy from children treated with the SIOPEN protocol for high-risk neuroblastoma. They hypothesize that this minimally invasive procedure could predict response to therapy and ultimately the prognosis of the patient.

The investigators identified three miRNAs that discriminated between good and poor responders to induction therapy. Total exosomes were significantly decreased by chemotherapy and the proportion of those derived from neuroblastoma (defined by GD2 positivity) significantly decreased after induction. These three miRNAs correlated with response and tumor growth.  The investigators also developed a chemoresistance index based on miRNA known to reflect resistance to chemotherapy for the individual drugs used in induction.  They separated two cohorts for this analysis depending on the miRNA vs drug profile.  The miRNA clusters for each cohort predicted overall prognosis of the subjects.

Questions and comments

Were the data regarding decreased exosomes after induction derived from the entire cohort of 52 subjects or just from 9 subjects Did the three identified miRNAs also predict EFS? To my naked eye, the two identified cohorts analyzed for chemoresistance appear to have a different proportion of good responders to bad responders. Is this correct and/or significant? How were the nine subjects analyzed for exosome content selected? Was the entire cohort analyzed only for chemoresistance? The investigators use miR and miRNA as abbreviations, chose one. I prefer miRNA Is the decrease in exosomes true down regulation or are the cells that secrete these miRNAs killed by chemotherapy and the remaining cells just do not?

Reviewer 4 Report

This is an well written manuscript describing a new exo-miR signature that may distinguish between which patients are poor versus good responders to Induction therapy. While the authors adequately express the need of new biomarkers of response in the high-risk neuroblastoma patient cohort, the manuscript as written falls short and requires further discussion of critical points and/or discussing the missing data points.

Introduction: The introduction, as written, is fine and adquatley describes the patient population and introduces the rationale behind the questions studied.

Methods: The authors need to better discuss where the 52 patients came from. I recognize that the patients were enrolled on the SIOPEN study; however, this is only a small subset of patients treated on this study. Were the samples selected in some manner that is not described? Was this during a specific time period and that is why the cohort number is smaller? The population need further discussion to ensure no bias was introduced in the selection of the patient population.

Results: I think it is peculiar that none of the patients reported in Table 1 had a CR to Induction therapy. Further none of the patients had PD to Induction therapy. This seems unusual. Additionally why did 5 patients have an end of Induction response of "N/A". Did these patients not undergo a disease evaluation due to progressive disease? These questions all need further explanation. As for measuring GD2 positivity, how was this measured? Was it only measured by flow? For the subset of patients (9) that were used to assess CD9 and GD2+ at baseline and at the end of Induction, how were these patients selected (random, based on response)? This needs to be further described for clarity. Also in the results section it describes that 6 patients had SD/MR and 8 had CR or VGPR; however, this is not consistent with what is reported in Table 1. Please clarify. 

Discussion: While I appreciate that there is a distinction in the signatures between the lowest responders and highest responders to Induction therapy, what happened to the patients (which is the majority since it includes 33 patients) who had an endo of Induction response of PR? The authors need to further discuss this large group of patients. Additionally, how can we incorporate this data? Presumably this information regarding response between MR/SD and VGPR/CR would be identified by imaging; did this analysis identify any patients that had a good response on imaging but a poor signature or vice versa. If there are a couple of examples that the authors could discuss (based on the patients overall outcome) that would make a much stronger case for future evaluations. This leads to my final comment which is why didn't the authors discuss these results in relation to relapse or outcome? Even reporting negative results is helpful to know that they performed the analysis. This is important information when discussing their results.

Round 2

Reviewer 4 Report

I think the authors addressed the concerns of the reviewers adequately. The manuscript is improved significantly over the first draft.